# Methylphenidate as a Novel Adjunct in Opioid-Taking Patients: Insights into Dopaminergic Neuroadaptation and Hypoactive Delirium

**DOI:** 10.3390/brainsci15080850

**Published:** 2025-08-08

**Authors:** Nikodem Świderski, Patryk Rodek, Krzysztof Kucia

**Affiliations:** 1Student Scientific Organisation of Department and Clinic of Adult Psychiatry, Faculty of Medical Sciences, Medical University of Silesia, Ziolowa 45, 40-635 Katowice, Poland; 2Department and Clinic of Adult Psychiatry, Faculty of Medical Sciences, Medical University of Silesia, Ziolowa 45, 40-635 Katowice, Poland; patrykrodek2208@gmail.com (P.R.);

**Keywords:** opioid use disorder, methylphenidate, dopaminergic neuroadaptation, opioid withdrawal, mesolimbic reward system

## Abstract

Background and aim of this review: The ongoing opioid epidemic underscores the urgent need for innovative pharmacological and behavioral interventions to mitigate the impact of opioid use disorder (OUD). This review aims to explore theoretical overlaps between the neurobiological mechanisms underlying OUD development and the pharmacodynamic profile of methylphenidate (MPH). Particular attention is given to the potential shared molecular targets, safety considerations, and therapeutic implications of MPH use in this clinical context. Main finding: In the development of opioid dependence, the negative reinforcement of the dopaminergic transmission of the mesocorticolimbic pathway induced by the supraspinal action of opioid receptor agonists plays a major role. The induced state of hypodopaminergic and hyperadrenergic modulates the underlying disease process by affecting cognitive control, affective regulation, and motivational drive. MPH, acting as a dopamine reuptake inhibitor and modulator of vesicular monoamine transporter 2 (VMAT-2), increases extracellular dopamine availability and enhances dopaminergic signaling, suggesting potential utility in restoring dopaminergic tone in OUD. Additionally, MPH has shown efficacy in hypoactive delirium in patients with terminal cancer, improving both cognitive function and psychomotor drive. Conclusions and future perspectives: There appear to be converging neurobiological mechanisms between the action of MPH and the pathophysiology of OUD, particularly within the dopaminergic system. However, well-designed clinical trials are essential to identify the patient subgroups that may benefit from adjunctive MPH treatment, to evaluate its efficacy in this setting, and to assess the long-term safety and risk profile of stimulant use in individuals with OUD.

## 1. Introduction

The misuse of opioid drugs has grown to be endemic in the United States, and many other countries are following the path set by the US (India, the United Kingdom and Canada) [1]. Opioid drugs are most commonly prescribed for the treatment of cancer pain, but are increasingly prescribed in therapies for chronic non-cancer related pain, despite controversies in terms of their efficacy, safety in long-term therapies, such as induced hyperalgesia, and the risk of misuse or the development of opioid use disorder (OUD) [2]. Intensive research is underway on new effective medications for opioid use disorder (MOUD), to better our understanding of the psychophysiology of the side effects of opioids in the CNS [3], as well as new potent analgetic compounds that do not exhibit addictive properties [4]. While MOUDs fulfill their role in OUD therapy, retention of the therapeutic value of treatment can be difficult. Numerous studies have demonstrated the roles of agonists (methadone), partial agonists (buprenorphine) and opioid receptor antagonists (extended-release naltrexone) in more effective OUD therapy [5]. Pharmacological treatment should be combined with behavioral therapy. The history of the opioid endemic in the U.S. probably dates back to 1999, when there was an increase in deaths from overdoses of opioid substances, mostly obtained from legal sources (natural opiates and semisynthetic opioids); the second wave of deaths mainly from heroin overdoses came around 2010, where the decreasing availability of prescribed drugs and the falling price of “street” heroin forced users to switch substances [6]. Currently, there is an ongoing upward trend in deaths from overdoses of fentanyl and fentanyl analogs from both legal and illicit sources. The increasing potency, addictive potential, and prevalence of opioid abuse, as well as the development of OUD, compels the search for new forms of pharmacological as well as behavioral therapy to combat the negative effects of addiction [7]. According to 2022 data, there are approximately 9.367 (95% confidence interval [CI] 8.603–10.196) million people over the age of 12 with active OUD, while 81,806 fatal opioid overdoses have occurred in the United States [8]. The economic burden created by opioid addiction and fatal overdose in 2017 was USD 1020.7 billion (95% [Cl] USD 967.2 to USD 1075.7 billion), about half of which is related to OUDs alone [USD 471.0 billion (95% [CI] USD 417.8 to USD 525.7 billion)] [9]. Meanwhile, the European Drug Report 2025 identified 8812 drug-related deaths across the European Union, Norway, and Turkey (EU+2) in 2023; nearly 70% of these fatalities involved opioids, either as the sole substance or in combination with other drugs [10].

This review presents a theoretical framework proposing the molecular substrates shared between the mechanism of action of MPH and the dopaminergic neuroadaptations that emerge during the development of OUD, highlighting potential overlaps in neuroplastic changes and compensatory mechanisms within the mesocorticolimbic system.

## 2. The Mechanism of Action of Opioids and the Neurobiological Basis of OUD Development

Opioids have found use in medicine as potent analgetics. Their analgesic action is based on three mechanisms—the inhibition of spinal cord projection neurons by small inhibitory interneurons, two descending pathways that have their origin in the periaqueductal gray matter, and the modulation of the supraspinal centers responsible for the psychosomatic dimension of pain [11]. Among non-classical opioids, tramadol and tapentadol stand out as atypical analgesics with dual mechanisms of action. Tramadol combines weak μ-opioid receptor agonism with the inhibition of serotonin and norepinephrine reuptake, and its active metabolite (O-desmethyltramadol) significantly contributes to its opioid activity [12]. However, its reliance on CYP2D6-mediated metabolism leads to interindividual variability and increases the risk of serotonin syndrome and seizures [13]. Tapentadol, in contrast, acts as a moderate μ-opioid receptor agonist and a potent norepinephrine reuptake inhibitor, without serotonergic involvement [14]. It does not require metabolic activation, offering more predictable pharmacokinetics and a potentially more favorable safety profile [15]. While both agents are considered to have lower abuse potential relative to strong opioids, they still carry a risk of dependence and misuse, particularly when used chronically or at high doses [16,17]. Notably, the combination of opioid and monoaminergic effects complicates withdrawal syndromes, which may include both classic opioid symptoms and features of noradrenergic or serotonergic rebound [18]. In addition to classical and non-classical opioids such as morphine, oxycodone, or fentanyl, there are growing concerns surrounding novel synthetic opioids (NSOs) such as nitazenes and designer fentanyl analogs, which often exhibit higher potency and lower detectability [19]. These substances do not always bind opioid receptors in a traditional manner and may possess atypical pharmacokinetic and pharmacodynamic profiles, contributing to a delayed onset of symptoms, prolonged effects, or reduced responsiveness to naloxone [20].

The affective action aiming to compensate for the emotional dimension of pain in the higher centers of the CNS involves the modulation of the administration of monoamines, especially dopamine and noradrenaline [21]. The opioid μ receptor agonist, through the inhibition of GABAergic interneurons, causes the disinhibition of dopaminergic ventral tegmental area (VTA) neurons belonging to the mesolimbic dopaminergic pathway. Thus, by reducing the inhibitory mechanisms, they increase cell firing and the amount of DA secreted through this pathway [22]. VTA dopaminergic neurons project to the nucleus accumbens (NAcc), which is part of the central reward system. Opioids, by implicitly stimulating the reward center, cause changes in decision-making when there is a conflict of interest. The stimulatory effect of the μ-opioid receptor agonist on VTA neurons and the stimulation of the reward system is thought to suppress the response to the noxious stimulus [23] (Figure 1). Dum and Herz [24] showed that priority over the pain stimulus is given to the pleasure stimulus. They conducted an experiment on rats in which they fed them on a hot plate. Of the two groups, the first was fed standard lab chow and the second was fed palatable chocolate treats. The control group jumped off the plate after an average of 5 s of feeding, while the one fed the treats spent an average of twice as long on the plate.

The stimulation of VTA dopaminergic neurons and locus coeruleus (LC) noradrenergic neurons by the regular, non-physiological input of opioid substances into the CNS provides a stimulus for the body to neuroadapt and consequently develop mechanisms to compensate for too-high dopamine concentrations in the reward system and enhanced LC neuron firing. Organic changes in the metabolism of monoamine neurotransmitters underlie withdrawal symptoms and provide the neurobiological basis for the occurrence of opioid use disorder (OUD) [25]. Opioids binding to MOP receptors inhibit the activity of noradrenergic neurons, resulting in a decrease in the secretion of this neurotransmitter [26]. Norepinephrine (NA) in the CNS stimulates responses related to wakefulness, breathing, blood pressure and general alertness, among other functions. The activated mu opioid receptor acts to inhibit neuronal adenylyl cyclase (AC1) [27]. During prolonged exposure to opioid substances, the available AC1 is increased in order to maintain proper NA excretion. However, when the ingested substance stops entering the CNS, LC neurons begin to produce large amounts of NA, causing withdrawal symptoms [28]. These include jitters, anxiety and muscle cramps.

Compensatory mechanisms that involve DA are mainly explained by two theories—“change set point” and “cognitive deficits” models. An organic change in the resting excitability of DA neurons leads to the inability of physiological factors to stimulate the central reward center, resulting in the need for a follow-up dose of a substance [29]. The set point excitability of dopaminergic neurons in the VTA is the result of two components—the excitatory action of cortical glutamatergic neurons and the inhibitory action of D2-autoreceptors [30]. Brake-type receptors are responsible for altering the resting dopamine concentration in the mesolimbic pathway that occurs in the persistence and development of OUD, with the task of reducing DA secretion by VTA neurons at too-high DA concentrations [31]. With frequent exposure to opioid substances, the CNS acts in opposition by increasing the number and strength of autobrake receptors [32]. In the event of opioid cessation, physiological factors are not powerful enough to generate an adequate response. The deprivation of DA concentrations manifests as dysphoria [33].

In addition to organic changes in the CNS, the development of OUD is linked to an emotional connection to the substance. In drug-less episodes, cue-induced craving has a plays a significant role in compulsive drug-seeking mechanisms [34]. The excitatory cortical neurotransmitters secreted during exposure to addiction cues stimulate neurons of the VTA regions to secrete DA and the locus coeruleus to secrete NA. The former has an effect on drug wanting or craving, while the latter enhances withdrawal symptoms [35]. Numerous studies indicate the significant influence that environmental factors and cognitive dysfunction induced by neuroadaptations forced by the consumption of a particular substance have on the mechanism of addiction [36]. The first of these is rooted in a person’s emotional dependence on the simple act of taking the drug, the place where he or she does it, or the people in close proximity. Cue-induced tolerance, which alters behavioral perceptions resulting from the use of a given substance, has been confirmed by numerous studies showing that an addict taking a standard dose of the drug of choice under new environmental conditions has a significantly higher chance of overdosing than if he or she did it as usual [37,38]. This sheds light on the environmental dimension of tolerance and its connection to emotional mechanisms in the drug–user relationship. Childress et al. [39] demonstrated an increase in limbic system neuronal activity by measuring cerebral blood flow using PET scans in cocaine-dependent patients, similar to that observed after cocaine use. The increased metabolism of neurons in the anterior cingulate, responsible for emotional expectations, and in the amygdala, responsible for emotional memory, suggest the emotional impact of craving, which may contribute to a better understanding of the mechanisms of addiction and thus to more effective treatment [40]. Both of the aforementioned centers are linked to the nucleus accumbens, which may imply that the mechanisms of craving are highly connected to the central reward system and, by stimulating them, are a factor modulating behavioral functions, which leads to the mechanism of compulsive drug seeking [41]. Cognitive functions and learning mechanisms are extremely important in the persistence of opioid use disorder (OUD) as well as the entire treatment process. How patients develop risk-increasing behaviors or drug-opposing responses is not entirely clear, but the mesolimbic dopaminergic pathway may be involved in both.

## 3. Pharmacokinetics and Pharmacodynamics of Methylphenidate

Methylphenidate (MPH) is a derivative of 2-piperidineacetic acid and exists in four isomers, thus having two stereogenic centers [42]. One pair of threo isomers and one erythro are distinguished, of which d-threo-methylphenidate shows the most expected clinical properties [43].

MPH is a psychostimulant and increases the activity of the central nervous system (CNS). The main mechanism of action is based on the inhibition of dopamine (DAT) and, to a lesser extent, norepinephrine (NET) transport proteins. Thus, MPH is a dopamine and norepinephrine reuptake inhibitor [44]. In addition, MPH binds to the serotonin transport protein (SERT) and serotonin receptors (5-HT_A1_ and 5-HT_B2_), with interactions distinguished by significantly high IC_50_ values and a lack of clinical relevance, (IC_50 (DAT)_ = 20 mM, IC_50 (NET)_ = 51 mM, IC_50 (5-HTA1)_ = 10,000 mM; for dl-MPH) [45] (Figure 2).

In addition, MPH selectively redistributes the vesicular monoamine transporter-2 (VMAT-2) [46] and alters pools of vesicles by reducing the quantity of vesicles to be recycled. Unlike amphetamine, which works by a similar mechanism, MPH induces the redistribution of vesicles from the plasmalemmal membrane-associated fraction to the cytoplasmic, non-membrane-associated fraction [47,48]. This reduces the risk of neurotoxicity and the accumulation of neurotransmitters in the cytoplasm of pericarion endings, which promotes the formation of reactive oxygen species [49]. VMAT-2 redistribution alters the dopaminergic transmission in neurons in a mechanism different from conventional DAT inhibition. Studies using PET showed that MPH has a higher affinity for DA transporters than cocaine and that the dose sufficient to inhibit 50% of DAT (ED_50_) is 0.25 mg/kg [50] by oral administration and 0.075 mg/kg by intravenous injection. The time required to produce a behavioral effect corresponds well with the peak uptake time of MPH in oral administration. Being an inhibitor of DA and NE transporters, MPH increases the concentration of these neurotransmitters in the synaptic cleft by preventing reuptake [51]. MPH increases the extracellular concentration of dopamine, mainly in the mesocortical and mesolimbic pathways, thus structurally in the striatum, nucleus accumbens and prefrontal cortex (PFC) [52]. In the case of norepinephrine, an increase is observed in the LC and in the PFC [53].

MPH is widely used as a first-line pharmacotherapy for attention-deficit/hyperactivity disorder (ADHD) in children and adults, and as a second-line treatment for narcolepsy [54,55]. In ADHD, MPH demonstrates robust efficacy in reducing core symptoms such as inattention, hyperactivity, and impulsivity by modulating frontostriatal dopaminergic and noradrenergic transmission, with effect sizes consistently reported in the moderate-to-large range [56]. While it is most frequently prescribed in childhood, MPH is increasingly used in adult ADHD, where it has shown significant benefits in improving executive functioning, emotional regulation, and occupational performance [57]. Beyond its approved indications, MPH has been explored in several off-label contexts. In depressive disorders, particularly treatment-resistant and apathy-dominant subtypes, MPH has shown potential as an adjunctive agent, particularly in the elderly or in medically complex patients where standard antidepressants pose risks [58,59]. Moreover, MPH has demonstrated utility in cancer-related fatigue, hypoactive delirium, post-stroke apathy, and traumatic brain injury, where the enhancement of catecholaminergic tone may facilitate cognitive activation, arousal, and goal-directed behavior [60,61,62].

MPH is generally well tolerated; however, dose-dependent adverse effects are frequently observed and include anorexia, insomnia, headache, irritability, and cardiovascular effects such as elevated heart rate and blood pressure [63]. Systematic reviews and meta-analyses report that up to 51% of subjects experience at least one non-serious side effect—most commonly decreased appetite (31%), sleep disturbances (18%), headache (14%), and abdominal pain (11%) [64]. Although randomized controlled trials generally show no statistically significant increase in serious adverse events compared to placebo in children and adolescents (RR = 0.98) [65], non-randomized cohort data suggest a modest increase in serious events (1.2% of treated individuals), including psychotic episodes and arrhythmias, which may occasionally necessitate treatment discontinuation [66]. In adult populations, isolated case reports have documented rare occurrences of reversible ischemic stroke, myocardial infarction, and psychosis [67,68,69].

Although MPH has a lower potential for abuse relative to amphetamine, it remains a controlled substance due to its reinforcement of dopaminergic activity. Nonmedical use is linked to euphoria, misuse, tolerance, and withdrawal symptoms such as fatigue, dysphoria, hypersomnia, increased appetite, and drug craving [70,71]. Withdrawal phenomena are typically milder than those observed with classic stimulants yet clinically significant in susceptible individuals.

## 4. Role of Methylphenidate in Stabilizing Dopaminergic Pathways

The main cause of withdrawal symptoms after opioid use relates to negative reinforcement, which forces neuroadaptations within brain reward circuitry or between the recruitment of brain stress circuitry. Within-system adaptations are attributed to changes triggered by the primary cellular response, which has its origin in the primary response of the reward system to eliminate substances and restore homeostasis [72]. Unlike the between-system response, where it is changes in other centers (antireward, stress, proinflammatory), changes occur as a result of primary changes in the reward system [73]. Within-system neuroadaptations involve the activation of G-proteins by MOP, which acting through secondary messengers can generate both short-term and long-term adaptations that are responsible for the tolerance process at the molecular and cellular level [74]. At the neuro circuit level, withdrawal symptoms are the result of a decrease in the extracellular dopamine concentration in the nucleus accumbens [75] and in the entire mesolimbic pathway through a decrease in DA neuron firing. Chronic morphine use also results in a decrease in the size of dopaminergic neurons in the ventral tegmental area and sensitization to dopamine receptor antagonists [76]. Changes in dopamine metabolism are also caused by an increase in GABA activity and metabotropic glutamine receptor activity. Both of these cause decreased cell firing in the ventral tegmental area [77]. Between-system adaptations involving norepinephrine, corticotropin-releasing factor (CRF), and dynorphin, among others, are the derivate factor responsible for inducing aversive feelings during opioid withdrawal and contribute to drug-seeking behaviors [78,79]. Chronic opioid administration disrupts the neurochemical state of homeostasis between systems that are located in the limbic area, specifically distinguishing the periaqueductal gray (PAG), amygdala, habenula and cingulate cortex, and affecting the connections of these areas in the dopaminergic neurons of the VTA and the central reward center [80]. The stimulation of areas with an inhibitory effect on cell firing within the VTA is the cause of withdrawal symptoms, but the source of aversive and dysphoric sensations is the mesolimbic dopaminergic pathway subjected to negative reinforcement.

Classical MOUD, including methadone and buprenorphine, act primarily through agonist or partial agonist activity at the μ-opioid receptor, reducing withdrawal symptoms and craving via substitution [81]. However, their mechanism does not directly address the underlying hypodopaminergic tone and frontostriatal dysregulation that contribute to cognitive impairment and motivational deficits in OUD. Dopaminergic enhancement induced by MPH may counteract opioid-induced anhedonia, apathy, and executive dysfunction, features inadequately addressed by conventional MOUD. Although theoretical and preclinical, this mechanistic divergence positions MPH as a candidate for augmentative therapy, particularly in individuals with persistent cognitive deficits, low motivation, or treatment-resistant profiles, provided that abuse liability is carefully managed.

## 5. Off-Label Prevention and Treatment of Opioids Withdrawal-Induced Hypoactive Delirium

Delirium is a neuropsychiatric disorder described by The Diagnostic and Statistical Manual of Mental Disorders, Fifth Edition (DSM-V) criteria as a sudden or fluctuating disturbance of consciousness and cognitive function that is caused by a physiological condition resulting from another disease entity, intoxication, or withdrawal [82]. Delirium is divided into subtypes based on the patient’s psychomotor functions, such as agitation, physical activity or sleep deprivation. Types include hyperactive, hypoactive and mixed delirium [83]. Patients with hypoactive delirium are largely inactive and exhibit significantly reduced alertness [84]. In addition, they may experience fear, incomprehensible experiences and strong feelings. Studies also show nightmares and a poor ability to rationalize experiences and assume their veracity [85]. Due to the difficult differential diagnosis (depression as well as dementia) of hypoactive delirium, the disease is often diagnosed late, which translates into a poorer prognosis and higher mortality [86]. There is compelling evidence that one of the factors inducing hypoactive delirium is the development of opioid withdrawal syndrome. However, it is less frequently diagnosed compared to other CNS-depressant withdrawal syndromes [87,88,89,90].

The neuropathogenesis of hypoactive delirium is not completely known, but recent reports favor the system integration failure hypothesis (SIFH) [91]. This assumes a disruption in the connectivity of the subcortical regions associated with cholinergic and dopaminergic conduction. Choi et al. constructed an experiment where using fMRI imaging, they confirmed a decrease in the connectivity of subcortical regions between the intralaminar thalamic nuclei, mesencephalic tegmentum, nucleus basalis, and ventral tegmental area both to a control sample and to measurements taken after the delirium had resolved [92]. The intralaminar thalamic nuclei and mesencephalic tegmentum show features of interconnection and are part of the ascending reticular activating system, which has roles in regulating arousal and consciousness [93]. The disruption of subcortical connections rich in dopaminergic neurons may suggest the influence of neuroadaptive processes occurring during opioid intake. The reduced activity of VTA dopaminergic neurons and, consequently, the entire mesocorticolimbic pathway can cause impaired transmission in those regions and predispose people to acute brain failure, manifested by delirium [94] (Figure 3). The inhibition of the level of activity of VTA dopaminergic neurons can manifest as interconnection disturbances between these systems, which supports the use of dopaminergic drugs in the prevention and treatment of delirium. MPH has been tested in the role of such pharmacological agents. It improved the motor retardation, slurred speech and alertness level of patients with hypoactive delirium who were in an advanced stage of cancer. Each patient showed an improvement in their Mini-Mental State Examination (MMSE) score, which increased from 21 to 27 (*p* < 0.001) on average after just one dose [95]. Thus, methylphenidate, which is a DAT inhibitor, needs to be clinically tested on a larger study group, being currently an off-label drug used in hypoactive delirium [96].

## 6. Discussion

The potential use of MPH in the treatment of OUD represents a novel and so far poorly studied area that deserves closer scientific attention. The aim of the present study was to identify common neurobiological mechanisms between the CNS stimulant mechanism of action (particularly those acting on the dopaminergic system, such as MPH) and opioid addiction, with a particular focus on alterations in reward circuitry, executive functions and neuroplasticity. The overlapping mechanisms of action between methylphenidate and OUD pathophysiology may provide a basis for considering its potential therapeutic use, although the currently available evidence is limited and mainly preclinical. Dysregulation within the mesocorticolimbic dopamine system plays a crucial role in opioid addiction. Chronic opioid use is associated with decreased dopamine receptor availability and the alteration of dopaminergic transmission, leading to anhedonia, a lack of motivation and compulsive substance-seeking behavior. MPH, as a dopamine and norepinephrine reuptake inhibitor, increases the concentration of catecholamines in the synaptic space, particularly in the prefrontal cortex and striatum [97]. Such action may compensate for the hypodopaminergic state observed in OUD, partially restoring reward sensitivity and executive control. MPH has been shown to improve cognitive functions such as attention, working memory and response inhibition in both healthy people and clinical patients, and these functions are frequently impaired in people with OUD [98,99].

Emerging data as well suggest that neurocognitive deficits, particularly in the scope of executive function and cognitive flexibility, may be a predictor of relapse in OUD [100]. The cognitive-enhancing properties of MPH may therefore promote treatment maintenance, increase engagement in psychotherapy and ultimately reduce the risk of relapse. Additionally, MPH may modulate impulsivity and reward deferral—two behavioral phenotypes frequently observed in OUD and associated with adverse treatment outcomes [101]. These points of commonality provide theoretical justification for exploring MPH as an adjunctive therapy in OUD, especially in patients with significant cognitive deficits or co-occurring attention deficit hyperactivity disorder (ADHD) [102].

Nevertheless, there are significant concerns. MPH is a psychostimulant with abuse potential, posing a risk of misuse or diversion, particularly in populations with a history of psychoactive substance use. Although extended-release formulations have less abuse potential than immediate-action forms, close monitoring and appropriate prescribing protocols would be essential [103]. Moreover, there is a lack of well-designed clinical trials evaluating the use of MPH directly in populations with OUD. Current evidence comes mainly from studies of related populations (e.g., people with ADHD and comorbid substance use alterations), or from experimental studies evaluating short-term dopaminergic effects rather than sustained clinical outcomes [104]. It is necessary to take into account the heterogeneity of OUD as well. Factors such as the duration and severity of addiction, the co-occurrence of other substance use, the presence of psychiatric alterations, and individual neurocognitive profiles could affect both the risks and potential benefits of MPH [105]. Personalized approaches, based on neuropsychological findings or brain imaging, could help identify the patients most likely to benefit from such an intervention.

From a mechanistic perspective, more research is needed on how MPH interacts with the endogenous opioid system. Although the dopamine system plays a crucial role in addiction, there are complex interactions between the opioid and monoaminergic systems—including shared signaling pathways, changes in receptor plasticity and epigenetic modifications—that are still not fully understood [106]. Animal models and neuroimaging studies in humans may help clarify whether MPH affects functional connectivity in the reward circuits impaired in OUD and whether this translates into clinically relevant effects.

## 7. Conclusions

In conclusion, although there is a solid neurobiological rationale for the further investigation of MPH in the context of OUD, the current state of knowledge does not allow for its routine clinical use. Well-designed randomized controlled trials are needed to assess efficacy, safety and optimal treatment parameters such as dosage, duration and patient selection. MPH may ultimately prove to be a useful therapeutic adjunct in a select group of patients with OUD—especially in people with cognitive deficits or ADHD—but its use must be based on sound empirical data and take into account potential risks. The intersection of stimulant therapy with opioid addiction treatment is a promising but complex area that deserves further research and careful clinical evaluation.


**Key finding:**
•The development and course of OUD is dependent on the neuroadaptive processes of the dopaminergic pathway of the mesocorticolimbic system.•These pathways are involved in cognitive and motivational processes and determine the propensity to relapse into addiction.•MPH, as a dopamine reuptake inhibitor and modulator of VMAT-2, may promote the tonic balance of dopaminergic pathways in the hypodopaminergic environment induced by opioid abuse.•Despite promising molecular correlations, well-designed clinical trials are needed to determine the preferred recipient group, the safety of adjuvant MPH therapy, and its potential benefits.•MPH has shown efficacy in reducing hypoactive delirium in patients with terminal cancer, which may share a clinical picture and potential pathogenesis with delirium induced by opioid withdrawal. The current state of knowledge does not allow for the routine use of MPH in this indication, whereas etiological similarities have prompted expanded clinical studies.


## Figures and Tables

**Figure 1 brainsci-15-00850-f001:**
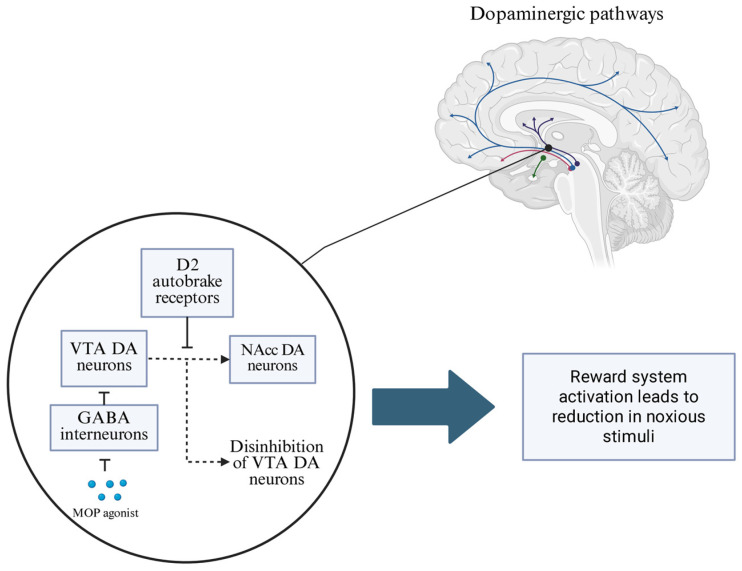
Activation of μ-opioid receptors in the ventral tegmental area inhibits local GABAergic interneurons, resulting in enhanced dopaminergic signaling to the nucleus accumbens. This dopaminergic modulation contributes to the affective blunting of noxious stimuli, reducing the emotional salience of pain. Additionally, increased dopamine levels engage presynaptic D2 autoreceptors, which provide negative reinforcement to regulate dopaminergic neuron excitability and maintain homeostasis within the reward system. (GABA—γ-aminobutyric acid; MOP agonist—μ-opioid receptor agonist; VTA DA neurons—ventral tegmental area dopaminergic neurons; NAcc DA neurons—nucleus accumbens dopaminergic neuron*s).*

**Figure 2 brainsci-15-00850-f002:**
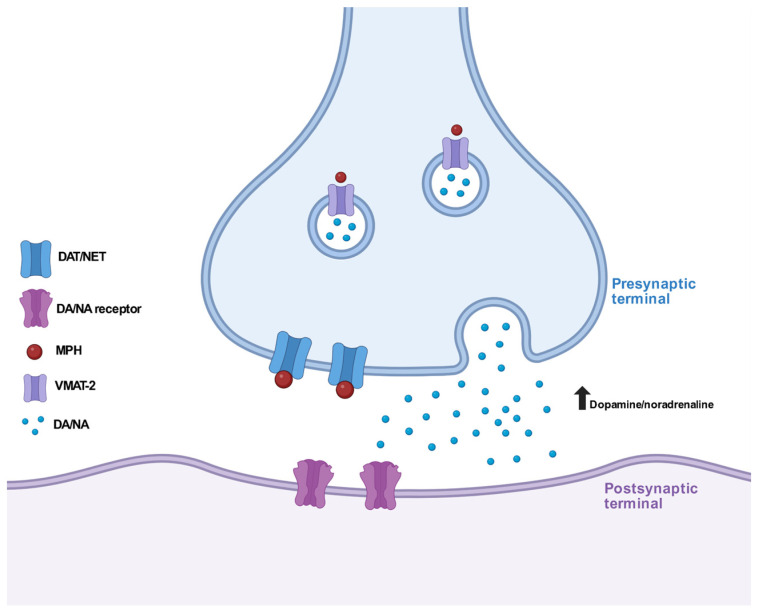
Methylphenidate inhibits dopamine and norepinephrine transporters, leading to increased extracellular concentrations of both neurotransmitters. Additionally, MPH modulates vesicular monoamine transporter 2, promoting the redistribution of synaptic vesicles and enhancing presynaptic dopamine release. (MPH—methylphenidate; DAT/NET—dopamine/norepinephrine transporter; DA/NE—dopamine/norepinephrine; VMAT-2—vesicular monoamine transporter-2; DA/NA receptor—dopamine/norepinephrine receptor*).*

**Figure 3 brainsci-15-00850-f003:**
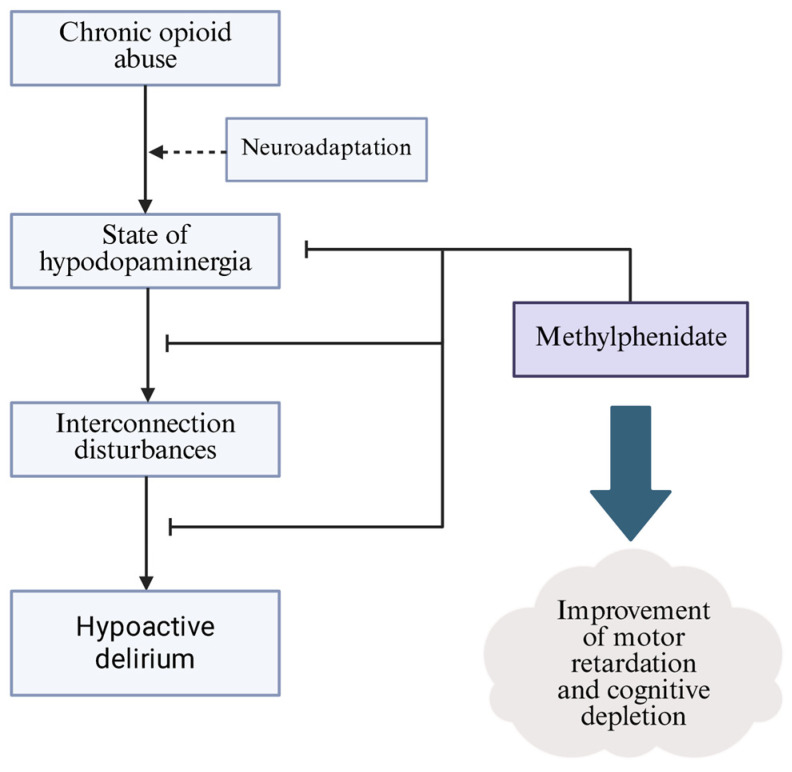
Schematic of the proposed pathogenesis of opioid withdrawal-induced delirium. MPH, through dopamine reuptake inhibition, can restore the activity of the mesocorticolimbic pathway, whose reduced activity has been observed in an episode of delirium.

## Data Availability

Additional data will be available on any inquiry in the form of e-mail correspondence author.

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
