# Peer review of "Methylphenidate as a Novel Adjunct in Opioid-Taking Patients: Insights into Dopaminergic Neuroadaptation and Hypoactive Delirium"

_brainsci, 2025, doi:10.3390/brainsci15080850_

Round 1
Reviewer 1 Report
Comments and Suggestions for Authors
The manuscript addresses the potential of methylphenidate as a promising adjunct in patients using opioids. This is an interesting topic, as methylphenidate may offer cognitive, analgesic or psychiatric benefits. Nevertheless, as the authors acknowledge, its use must be cautious and individually assessed. However, several aspects require clarification and improvement to enhance the manuscript’s scientific robustness and overall clarity. Additionally, the manuscript would greatly benefit from the inclusion of a table or figure summarizing key concepts. Such an element would help integrate and synthesize the main points discussed.
I believe this paper holds potential relevance, fits the scope of Brain Sciences, and could be considered for publication once the concerns listed below are addressed:
- The Abstract section could be reformulated. In addition to contextualizing the topic, it should clearly state the aim of the review, highlight the main conclusions, and outline the potential impact or future perspectives.
- The authors could be considered add new keywords, such as: dopaminergic neuroadaptation, opioid withdrawal, monoamine metabolism, dopaminergic pathways, norepinephrinergic system or mesolimbic reward system.
- The abbreviation "OUD" should be written out in full (opioid use disorder) at its first occurrence in the main text (in the introduction section), even though it already appears in the Abstract.
- In the Introduction, besides the quantitative data regarding the United States, it would be useful to include statistics from other regions to demonstrate the global relevance of the topic.
- The Introduction should end with a clear statement of the aim of the present review.
- Figure legends: To improve clarity, figure legends should define all abbreviations in full and provide more comprehensive descriptions that guide the reader in understanding the visual information.
- Page 4, line 111: “Page 6 of 18” - Is this information necessary, or is it a formatting error?
- Several text segments lack references. Please ensure that all scientific claims are properly supported by literature.
- Regarding section “2. The mechanism of action of opioids and the neurobiological basis of OUD development”, it would be interesting to compare classical with atypical opioids, such as tramadol, which acts not only as a µ-opioid receptor agonist but also as a norepinephrine and serotonin reuptake inhibitor. Its use and misuse are currently recognized as a global issue.
- Certain aspects concerning methylphenidate could be further developed, as: clinical uses, adverse effects, and abuse potential.
- The authors might consider including a summary figure or table relating opioid and/or methylphenidate use with dopamine levels and the potential link to delirium.
Author Response
We would like to thank the reviewers for their insightful and constructive comments. We appreciate the time and effort dedicated to reviewing our manuscript and believe that the suggested revisions have substantially improved the quality and clarity of the work. Below, we address each comment point-by-point:
Comment 1: "Additionally, the manuscript would greatly benefit from the inclusion of a table or figure summarizing key concepts."
Response 1: Thank you for the valuable suggestion. A summary of key findings from this review has appeared in the “Conclussion” section. [lines 390-405]
Comment 2: "The Abstract section could be reformulated. In addition to contextualizing the topic, it should clearly state the aim of the review, highlight the main conclusions, and outline the potential impact or future perspectives."
Response 2: There has been a restructuring of the abstract with a clear distinction between the purpose of this review, a systematization of the knowledge achieved, and a summary and outlook for future research. [lines 11-30]
Comment 3: "The authors could be considered add new keywords, such as: dopaminergic neuroadaptation, opioid withdrawal, monoamine metabolism, dopaminergic pathways, norepinephrinergic system or mesolimbic reward system."
Response 3: New keywords have been added as suggested, currently consisting of phrases: "opioid use disorder; methylphenidate; dopaminergic neuroadaptation; opioid withdrawal; mesolimbic reward system; opioid withdrawal-induced delirium;"
Comment 4: "The abbreviation "OUD" should be written out in full (opioid use disorder) at its first occurrence in the main text (in the introduction section), even though it already appears in the Abstract."
Response 4: The acronym has been expanded as suggested. [line 40]
Comment 5: "In the Introduction, besides the quantitative data regarding the United States, it would be useful to include statistics from other regions to demonstrate the global relevance of the topic."
Response 5: Data on drug-related deaths from the European Drug 2025 report covering the entire European Union, Norway and Turkey have been added. [lines 62-65]
Comment 6: "The Introduction should end with a clear statement of the aim of the present review."
Response 6: The statement has been added to clearly indicate the purpose of this review. [lines 66-69]
Comment 7: "Figure legends: To improve clarity, figure legends should define all abbreviations in full and provide more comprehensive descriptions that guide the reader in understanding the visual information."
Response 7: The descriptions of the figures have been rewritten to make it easier to understand the visual content depicted on them. Expansions of all abbreviations on them have been added as well. [Fig 1.: lines 113-120; Fig 2.: lines 191-196; Fig 3.: lines 327-329]
Comment 8: "Page 4, line 111: “Page 6 of 18” - Is this information necessary, or is it a formatting error?"
Response 8: The formatting error has been removed and the article has been checked again for their occurrence.
Comment 9: "Several text segments lack references. Please ensure that all scientific claims are properly supported by literature."
Response 9: The text was supplemented with references for every place where one was missing, the total number of references increased from 67 to 103. (Some were added in the existing text, whereas the rest came from new paragraphs included in the manuscript, added after review report analysis.)
Comment 10: "Regarding section “2. The mechanism of action of opioids and the neurobiological basis of OUD development”, it would be interesting to compare classical with atypical opioids, such as tramadol, which acts not only as a µ-opioid receptor agonist but also as a norepinephrine and serotonin reuptake inhibitor. Its use and misuse are currently recognized as a global issue."
Response 10: A description of non-classical opioids (tramadol, tapentadol) has been added, along with the mechanism of action, risk of addiction and basic pharmacological profile. Information on new synthetic opioid substances (NSOs) has been added as well, along with a brief description. [lines 76-95]
Comment 11: "Certain aspects concerning methylphenidate could be further developed, as: clinical uses, adverse effects, and abuse potential."
Response 11: The paragraph describing methylphenidate has been expanded to include the profile of adverse reactions (along with frequency in the adult population), clinical indications and also off-label uses, and the potential for addiction. [lines 214-246]
Comment 12: "The authors might consider including a summary figure or table relating opioid and/or methylphenidate use with dopamine levels and the potential link to delirium."
Response 12: Neurotransmitter levels during an episode of delirium are reported by sources as variable, as well as lacking reliable data on the direct correlation of levels of a given neurotransmitter and onset of delirium. We considered misleading information about baseline dopamine levels before and during an episode of delirium, receiving the leading SIFH pathogenesis hypothesis (system integration failure hypothesis), which describes the pathomechanism at the connectome level. The description included information about the effect of methylphenidate on the described regions both at the molecular level and the neuronal networks.
Reviewer 2 Report
Comments and Suggestions for Authors
The review manuscript by Nikodem Åšwiderski and co-authors presents a theoretical framework exploring the potential role of methylphenidate (MPH) as an adjunctive treatment for opioid use disorder (OUD) and opioid withdrawal-induced hypoactive delirium. The manuscript provides an overview of the neurobiological mechanisms underlying opioid addiction, with particular focus on dopaminergic and noradrenergic pathways. The authors propose a rationale for potential therapeutic application of MPH in this context. The review will be interesting to the journal readership and deserves publication.
Specific comments:
1) Lines 138-139: Methylphenidate (MPH, CAS 20748-11-2) lacks both the benzopiperidine and phenylethylamine structural scaffolds as substructures. Thus, it cannot be classified as a derivative of either. Correctly, compound MPH should be called a derivative of 2-piperidineacetic acid. See for reference:
https://commonchemistry.cas.org/detail?cas_rn=20748-11-2
2) Lines 229-230: The reference to citations [54, 55] appears to be out of sequence with the surrounding text. Please check also throughout the manuscript.
3) The manuscript would benefit from a more pronounces comparison of the MPH's theoretical advantages against other potential pharmacological approaches for OUD treatment.
Summarizing, I recommend minor revision of the manuscript before acceptance.
Author Response
We are grateful for the thoughtful and detailed feedback provided by the reviewers. We have carefully considered all the suggestions and revised the manuscript accordingly. Please find our point-by-point responses below.
Comment 1: "Lines 138-139: Methylphenidate (MPH, CAS 20748-11-2) lacks both the benzopiperidine and phenylethylamine structural scaffolds as substructures. Thus, it cannot be classified as a derivative of either. Correctly, compound MPH should be called a derivative of 2-piperidineacetic acid. See for reference: https://commonchemistry.cas.org/detail?cas_rn=20748-11-2 "
Response 1: Thank you for your valuable suggestion, the name has been corrected and the sentence rephrased: "Methylphenidate (MPH) is a derivative of of 2-piperidineacetic acid and exists in 4 isomers, thus has 2 stereogenic centers." [lines 179-180]
Comment 2: "Lines 229-230: The reference to citations [54, 55] appears to be out of sequence with the surrounding text. Please check also throughout the manuscript."
Response 2: The attributed citations have been corrected, as well as new ones have been added to support some sections. Their total number has increased from 67 to 103 (some of the added ones are from new fragments suggested by the second reviewer).
Comment 3: "The manuscript would benefit from a more pronounces comparison of the MPH's theoretical advantages against other potential pharmacological approaches for OUD treatment."
Response 3: Thank you for the suggestion. A section has been added comparing conventional drugs used in OUD (mu opioid rec. agonists and partial agonists) with the theoretical action of methylphenidate. The differences in the target site of action and the mechanism in which the drug is involved in the therapeutic process have been detailed
We would like to once again thank the reviewers and editors for their thorough and constructive evaluation of our manuscript. We believe that the revisions made in response to the comments have significantly improved the clarity, scientific rigor, and overall quality of the paper. We hope that the updated version meets the expectations of the journal and look forward to your further feedback.